A predictive framework for identifying source populations of non-native marine macroalgae: Chondria tumulosa in the Pacific Ocean

Fumo James T. 1 jfumo@hawaii.edu
http://orcid.org/0000-0002-2772-4709 Nichols Patrick K. 1
Ely Taylor 1
http://orcid.org/0000-0002-2061-5841 Marko Peter B. 1
Moran Amy L. 1
Powell Brian S. 2
http://orcid.org/0000-0002-8340-7804 Williams Taylor M. 3
http://orcid.org/0000-0003-1363-5702 Kosaki Randall K. 4 5
http://orcid.org/0000-0002-5794-8973 Smith Celia M. 1
Lopes Keolohilani H. Jr. 6
Smith Jennifer E. 7
Spalding Heather L. 8
http://orcid.org/0000-0002-7324-7448 Krueger-Hadfield Stacy A. 3 9
http://orcid.org/0000-0002-7663-6545 McDermid Karla J. 10
Hauk Brian B. 5
Morioka James 11
O’Brien Kevin 11
Kennedy Barbara 12
http://orcid.org/0000-0002-4627-7318 Leliaert Frederik 13
Fujii Mutue T. 14
Nelson Wendy A. 15 16
Draisma Stefano G. A. 17
Sherwood Alison R. 1
1 School of Life Sciences, University of Hawai‘i at Mānoa , Honolulu, Hawai‘i , United States
2 Department of Oceanography, School of Ocean and Earth Science and Technology, University of Hawai‘i at Mānoa , Honolulu, Hawai‘i , United States
3 Department of Biology, University of Alabama—Birmingham , Birmingham, Alabama , United States
4 Center for the Exploration of Coral Reef Ecosystems (XCoRE), Bernice Pauahi Bishop Museum , Honolulu, Hawai‘i , United States
5 National Oceanic and Atmospheric Administration , Honolulu, Hawai‘i , United States
6 Department of Natural Resources and Environmental Management, University of Hawai‘i at Mānoa , Honolulu, Hawai‘i , United States
7 Scripps Institution of Oceanography, University of California, San Diego , La Jolla, California , United States
8 Department of Biology, College of Charleston , Charleston, South Carolina , United States
9 Virginia Institute of Marine Science Eastern Shore Laboratory, College of William and Mary , Wachapreague, Virginia , United States
10 Marine Science Department, University of Hawai‘i at Hilo , Hilo, Hawai‘i , United States
11 Papahānaumokuākea Marine Debris Project , Kailua, Hawai‘i , United States
12 Herbarium Pacificum, Bernice Pauahi Bishop Museum , Honolulu, Hawai‘i , United States
13 Meise Botanic Garden , Meise , Belgium
14 Biodiversity Conservation Center, Environmental Research Institute , São Paulo , Brazil
15 Tāmaki Paenga Hira Auckland Museum , Auckland , New Zealand
16 School of Biological Sciences, University of Auckland , Auckland , New Zealand
17 Excellence Center for Biodiversity of Peninsular Thailand, Prince of Songkla University , Hat Yai , Thailand
Zhukova Natalia
Electronic publication date: 2025 Jun 23
Publication date: 2025
Volume: 13
Electronic Location ID: e19610
Received 2025 Mar 17; Accepted 2025 May 26
Copyright: © 2025 Fumo et al.
Copyright year: 2025
Copyright holder: Fumo et al.
License: This is an open access article distributed under the terms of the Creative Commons Attribution License, which permits unrestricted use, distribution, reproduction and adaptation in any medium and for any purpose provided that it is properly attributed. For attribution, the original author(s), title, publication source (PeerJ) and either DOI or URL of the article must be cited.
License URL: https://creativecommons.org/licenses/by/4.0/

Keywords: Dispersal, Cryptogenic, Algae, eDNA, Connectivity modeling, Herbaria, Barcoding, Backtracking

Funding: United States National Fish and Wildlife Foundation (NFWF) #0810.22.074191 Achievement Reward for College Scientists (ARCS) Foundation NFWF Foundation Papahānaumokuākea Research and Conservation Fund #0810.20.074235 National Science Foundation Division of Ocean Sciences (NSF-OCE) #2049673 Barcoding of herbarium specimens was supported by a grant secured by Alison Sherwood from the United States National Fish and Wildlife Foundation (NFWF) award #0810.22.074191. Additional support was received through an Achievement Reward for College Scientists (ARCS) Foundation award to James Fumo. Attendance at the Chondria Conference (ChonCon) was supported by NFWF Foundation Papahānaumokuākea Research and Conservation Fund award #0810.20.074235 to Heather Spalding and Stacy Krueger-Hadfield. Field eDNA sampling at Johnston, Kiribati, and Okinawa was funded by National Science Foundation Division of Ocean Sciences (NSF-OCE) award #2049673 to Peter Marko and Amy Moran. The funders had no role in study design, data collection and analysis, decision to publish, or preparation of the manuscript.

==============================
The cryptogenic marine red alga Chondria tumulosa was first observed in 2016 in subtidal habitats at Manawai (Pearl and Hermes Atoll) in the Papahānaumokuākea Marine National Monument (PMNM), Hawai‘i. Without molecular or morphological matches to any known species, it was described in 2020 and declared cryptogenic. This alga has substantially increased in benthic cover and has been discovered on two additional atolls in PMNM: Kuaihelani (Midway) and Hōlanikū (Kure). It exhibits several characteristics indicative of non-native origins including putative prior absence in the region, persistence in high densities over nearly a decade, apparent lack of native herbivore pressure, and strong tetrasporophytic bias. Importantly, it is negatively impacting the culturally and ecologically valuable reefs of PMNM. The geographical origin of this putative invasion is unknown, and there are no published reports of the species occurring anywhere other than PMNM. The central Pacific location of Hawai‘i allows a broad range of potential sources for the origin of C. tumulosa. Taxonomic ambiguities within the genus Chondria and challenges associated with sampling necessitate the development of a narrowed set of search locations and efficient search strategies to detect the species outside of PMNM. Attachment to floating debris is a potential introduction vector for C. tumulosa into PMNM, and an oceanographic model was used to identify the most likely source locations for this pathway between 2000 and 2015, including Japan in the western Pacific, Johnston Atoll, the Line Islands including Palmyra Atoll in the central Pacific, and Clipperton Atoll and the Galápagos Islands in the eastern Pacific. We used a recently developed and validated eDNA assay for detecting C. tumulosa from three of the regions of interest to screen for C. tumulosa with no samples yielding positive detections. We provide a framework for investigating positive eDNA field detections using in-water surveys, microscopy, and DNA barcoding. A parallel sampling effort targeting preserved specimens stored in global herbaria is also presented, which did not yield any detections. Several Chondria species remain targets for sequencing from global herbaria. Identification of the native range of C. tumulosa is a critical step that will allow for an evaluation of its evolutionary ecology and any shifts that may have occurred that facilitated its putative invasion and subsequent spread, offering insights crucial for the development of mitigation strategies to safeguard PMNM against further risk.

Introduction

Papahānaumokuākea Marine National Monument (PMNM) is a protected marine region of over 1,500,000 km2 surrounding the atolls and islands to the northwest of the populated Main Hawaiian Islands (MHI). It holds substantial value as a part of the cultural heritage of Hawai‘i with cosmological importance and symbolic meaning to Native Hawaiians (Kosaki, Chow & Keenan, 2009; Wiener & Wagner, 2013; Gaymer et al., 2014; Kikiloi et al., 2017). PMNM was designated as a World Heritage Site in 2010 by the United Nations’ Education, Scientific and Cultural Organization (UNESCO), and recognized as the first mixed site in the United States for its scientific and cultural importance (Abdulla et al., 2013). It is co-managed by several entities, including the State of Hawai‘i, the U.S. Fish and Wildlife Service, the National Oceanic and Atmospheric Administration (NOAA), and the Office of Hawaiian Affairs with an incorporation of Native Hawaiian values, serving as an example for other large scale protected areas globally (Toonen et al., 2013; Kikiloi et al., 2017).

Legal protection and the remoteness of PMNM have helped preserve this region, which is relatively pristine in comparison to the MHI and other populated areas worldwide, and is largely shielded from direct anthropogenic stressors, such as overfishing, eutrophication, sedimentation, and the risk of non-native species introductions (Selkoe et al., 2009). Fish communities of PMNM are dominated by apex predators (Friedlander & DeMartini, 2002; Holzwarth et al., 2006) and its atolls are characterized by low rates of coral disease (Kenyon et al., 2006), healthy benthic communities (Vroom & Braun, 2010), and large populations of birds and turtles (Harrison, 1990; Balazs & Chaloupka, 2004; Dale, Meyer & Clark, 2011). The relatively pristine state of PMNM provides a baseline for the study of coral reef ecosystems globally, and those in the MHI. For example, the relative dominance of macroalgae in PMNM diverges from the paradigm of coral dominance in “healthy” reef systems, indicating that strategies for reef health assessment need revision (Vroom & Braun, 2010). Despite the overall robust health and relatively pristine condition of PMNM reefs, the region is not immune to global challenges, such as climate change, sea level rise, and marine debris (Selkoe et al., 2009; Royer et al., 2023). The emergence of novel or previously undocumented taxa demonstrates the need to better understand the biogeographic origins of species in Hawai‘i.

Indo-Pacific biogeographic patterns in algae remain an enigma. The historical biogeography of the brown alga Lobophora J. Agardh (Vieira et al., 2017) and the green algal family Udoteaceae J. Agardh (Lagourgue, Leliaert & Payri, 2022) support patterns of dispersal from the western Indo-Pacific to the central Pacific islands, followed by limited diversification. One of the few studies to examine red algal phylogeography across the Indo-Pacific focused on the genus Portieria Zanardini (Leliaert et al., 2018). The findings suggest that diversity in Hawai‘i may have been imported from the southwestern Pacific (i.e., Tonga, Fiji, New Caledonia, and the east coast of Australia). Additionally, diversity within the Hawaiian archipelago appears to be relatively low in comparison to the rest of the sampled regions in the Indo-Pacific (Leliaert et al., 2018). Further, the study indicates there is no evidence of export of Portieria diversity from Hawai‘i back into the Indo-Pacific (Leliaert et al., 2018). Red algal propagules are short lived (Hoffmann & Camus, 1989), negatively buoyant (Okuda & Neushul, 1981; Hoffmann & Camus, 1989), and attach rapidly to nearby substrata when released (Okuda & Neushul, 1981; Fletcher & Callow, 1992). Moreover, male gametes are non-flagellated (Santelices, 1990) while female gametes do not disperse and are retained on the female gametophyte (Searles, 1980). Thus, red algae are unlikely to disperse via spores or gametes over long distances, exhibiting patterns of isolation by distance from small to large spatial scales (e.g., Engel, Destombe & Valero, 2004; Krueger-Hadfield et al., 2013, 2017).

While information regarding the biogeography and phylogeography of marine macroalgae in Hawai‘i is limited, the origin of Hawai‘i’s marine animal biodiversity has been studied in some detail (Briggs & Bowen, 2012, 2013). Biogeographic connectivity of Hawai‘i’s marine fauna is mainly with the Indo-Polynesian province (Briggs & Bowen, 2012) where intermediary islands including Johnston Atoll and the Line Islands can act as stepping stones (Bowen et al., 2013). Connectivity also exists with Japan in the western Pacific via the Kuroshio Current and the North Pacific Subtropical Gyre (Friedlander et al., 2009). Fewer connections exist between the central Pacific and the Eastern Tropical Pacific (ETP) due to the large open ocean distances separating the two regions (Briggs & Bowen, 2013). Occasional connections across this soft boundary do occur in animals and these events disperse genetic material in both directions (Lessios & Robertson, 2006; Wood et al., 2016; Romero-Torres et al., 2018).

Marine debris facilitates the dispersal of organisms, overcoming natural dispersal barriers and connecting regions of distinct biodiversity. Entanglement in human-made marine debris enables essentially indefinite dispersal for certain organisms provided their physiological requirements for light, nutrients, and temperature are met (Simkanin et al., 2019). Marine debris is increasingly recognized as a vector for introductions, and has the potential to facilitate transit across exceptional distances from source to sink locations (Carlton et al., 2018; Chong et al., 2023; Haram et al., 2023; Carlton & Schwindt, 2024). The dispersal of organisms on marine debris was demonstrated by the 2011 Tōhoku earthquake, which generated substantial amounts of material that rafted to the Hawaiian Islands carrying living marine organisms that dispersed as far as the Oregon coast (Carlton et al., 2017, 2018). Further, PMNM is impacted by an abundance of large, floating, abandoned, lost, or discarded fishing gear, which acts as a habitat substitute in the open ocean with the capacity for non-native species transport and introduction (Coleman et al., 2014; Royer et al., 2023; Benadon et al., 2024). Manawai Atoll (Pearl and Hermes Atoll) leads the PMNM in volume of marine debris, with 505 metric tons removed over the last 40 years (Baker et al., 2024). Given the dispersal limitations of Rhodophyta, marine debris—alongside hull fouling and ballast water from both commercial and recreational vessels—may be vectors of importance in moving red algae through the Pacific Ocean.

Chondria tumulosa Sherwood and Huisman, first observed at Manawai in 2016 (Sherwood et al., 2020), is a red alga that has been negatively impacting reefs of the northernmost atolls of PMNM for nearly a decade (Sherwood et al., 2020; Lopes et al., 2023). Lacking a morphological or DNA barcoding sequence match in any publicly available database, a new species was described and declared cryptogenic (Sherwood et al., 2020). While the first observation of C. tumulosa at Manawai was in 2016 (Sherwood et al., 2020), satellite observations of accumulations of fragmented clumps revealed the species was likely present since at least 2015 (Lopes et al., 2023). Spatial analysis of these accumulations showed an 115-fold increase in surface area coverage from 2015–2021 at Manawai (Lopes et al., 2023). In 2021, the alga was observed ~150 km to the northwest of Manawai at Kuaihelani (Midway Atoll) and in 2023 it was observed at Hōlanikū (Kure Atoll), which is located ~100 km to the northwest of Kuaihelani and the northwestern-most atoll in PMNM. Entangled thalli form mats up to 18 cm thick, which smother the existing benthic subtidal community in patches of several thousand square meters (Sherwood et al., 2020). Mats have been observed attached, loosely adhered, tumbling along the benthos, and acting as flotsam in PMNM (Sherwood et al., 2020; Lopes et al., 2023). The prolific growth of these mats is a threat to the biodiversity and abundance of native benthic species that are overgrown. Mats or clumps of the alga break away and disperse along the benthos, and more rarely disperse along the surface of the ocean; presumably, these thallus fragments can establish and grow in new areas via secondary reattachments and asexual processes (Sherwood et al., 2020; Lopes et al., 2023; Williams et al., 2024). Accumulations of fragmented clumps, which tend to be negatively buoyant, gather in regions of bathymetric depressions and form dark patches visible from high resolution satellites (Lopes et al., 2023). Moreover, the sites sampled at Manawai in 2019 were dominated by tetrasporophytes—a hallmark of range expansions and asexual reproduction in haploid-diploid algae (Krueger-Hadfield, 2020)—as no gametophytic thalli were observed using microscopy or microsatellite genotyping and there are genetic signatures of asexual reproduction (e.g., repeated genotypes) (Williams et al., 2024). Observations from authors during in-water surveys suggest growth forms of C. tumulosa can vary from robust and mat-forming to cryptic and epiphytic.

Although C. tumulosa is currently considered cryptogenic (sensu Carlton, 1996), there are multiple indications that the species is recently introduced to PMNM. The growth habit of C. tumulosa (Sherwood et al., 2020), its persistence in high densities over nearly a decade (Lopes et al., 2023), tetrasporophytic dominance facilitated by asexual thallus fragmentation (Williams et al., 2024), signatures of clonality not expected in a native species (Williams et al., 2024), putative prior absence in the region, a distribution restricted to three atolls (Nichols et al., 2025), anthropogenic dispersal potential (Lopes et al., 2023; Fumo et al., 2024), as well as its apparent lack of native herbivore pressure (Lopes et al., 2023), are consistent with the possibility that C. tumulosa is a recent introduction to PMNM (see also Carlton & Schwindt, 2024). Additionally, the species has been observed rafting on flotsam in PMNM (Lopes et al., 2023), indicating that this may be the dispersal vector not only for the recent spread of the species within PMNM (Fumo et al., 2024), but perhaps also into the region from elsewhere in the Pacific Ocean. As such, while there are other potential means of introduction—including ballast and hull fouling by both recreational and commercial vessels—drift material may be considered the most plausible vector for the introduction of C. tumulosa to PMNM.

Observations of trans-oceanic rafting events, including those of C. tumulosa in PMNM (Lopes et al., 2023), demonstrate that taxa once confined to their respective biogeographic zones can overcome long established dispersal barriers via episodic rafting events (Thiel & Haye, 2006; Carlton et al., 2017; Benadon et al., 2024). Further, species can establish, persist, and spread into novel environmental conditions, complicating efforts to pinpoint their origins. For example, the prolific red algal invader Gracilaria vermiculophylla (Ohmi) Papenfuss occurs in warmer waters throughout its non-native range compared to its source populations (Sotka et al., 2018). Thus, because the potential source regions of C. tumulosa are broad, researchers must develop strategic and efficient mechanisms for sampling regions of interest.

Given the broad range of potential origins and varied pathways by which C. tumulosa could have arrived in PMNM via drift material, connectivity modeling may help identify key areas to prioritize in sampling efforts and enable more efficient and targeted searches. By integrating environmental and biological factors, these models help predict dispersal and connectivity in marine systems (Vaz, 2012; Wren et al., 2016; Swearer et al., 2019; Hixon et al., 2022; Fumo et al., 2024) and are validated by congruence between modeled and observed connectivity (e.g., Fraser et al., 2022; Counsell et al., 2022). Moreover, connectivity modeling helps in the examination of the spread and arrival of invasive species (Treml et al., 2008; Johnston & Purkis, 2011, 2013; Gabriel et al., 2024) and serves as a cost-effective tool for conservation (Schill et al., 2015) and connectivity (Swearer et al., 2019), contributing to an integrated understanding of species dispersal in marine environments.

Accurate and sensitive detection methods are crucial for understanding the dispersal of nuisance taxa across oceanic basins; however, the resources required to visually survey all potential source locations render this option unfeasible. Instead, molecular detection methods, such as environmental DNA (eDNA)—any genetic material obtained from the environment—may be better suited for rapid and widespread detection of rare aquatic taxa (Darling & Blum, 2007; Goldberg et al., 2013; Doyle, McKinnon & Uthicke, 2017; Mauvisseau et al., 2018; Keller et al., 2022; Gargan et al., 2022). By using species-specific molecular assays, eDNA detection can greatly increase sensitivity over conventional direct observation methods (Harvey, Hoy & Rodriguez, 2009; Smart et al., 2015; Burian et al., 2021; Keller et al., 2022). When target DNA concentrations exceed established thresholds for detection and quantification (Klymus et al., 2020), eDNA can also provide information on relative abundance and help guide more intensive non-molecular surveys (Jerde et al., 2011). Recently, an assay has been developed and validated for detecting C. tumulosa in Hawai‘i, exhibiting high sensitivity even at low abundances in shallow reef environments (Nichols et al., 2025).

The morphological and genetic complexities of distinguishing species of Chondria and related genera highlight the crucial role molecular analysis plays in confirming species identity and distributional range. There are currently 78 accepted species names for Chondria, while 69 have been placed in synonymy (Guiry & Guiry, 2024). When including all species that are accepted taxonomically, of unresolved taxonomic status, or have been transferred to other genera within the family Rhodomelaceae, there are 113 species (Guiry & Guiry, 2024). Chondria tumulosa is morphologically distinct from other species in the genus in having large, robust thalli, a mat-forming tendency, and terete, bluntly rounded axes with decreasing diameter with each level of branching—morphological characteristics maintained when the species occurs both in high densities and cryptically (Sherwood et al., 2020). The mat-forming habit of the species is a result of multicellular haptera attaching and reattaching the thallus to the substratum, and a gradient of colors exists from golden-brown to dark brown or purple from the upper to lower sections of the mat where irradiance is limited by self-shading (Sherwood et al., 2020). The Universal Plastid Amplicon (UPA) provides a useful DNA barcode because it exhibits high amplification and sequencing success rates from degraded DNA; however, it is a relatively conserved region in comparison to other commonly used markers (Sherwood et al., 2010; Gabriel et al., 2020). Despite the conserved nature of UPA, no published matches have yet been identified for C. tumulosa outside of PMNM, implying this barcode can be used for initial screening of specimens. However, as many species of Chondria have yet to be sequenced using the UPA barcode, there may be species which share exceptionally similar sequences. Thus, further sequencing of the nuclear SSU rDNA, mitochondrial COI, and plastidial rbcL DNA barcodes can be used to bolster identifications of C. tumulosa (Sherwood et al., 2020). The full plastid genome of C. tumulosa is also available on GenBank under accession MW309501 (Paiano et al., 2021).

The goals of this study were to use a combination of the techniques to better understand the likely origin of this species. Specifically, we (i) narrow the scope of potential source locations of C. tumulosa using spatial modeling, (ii) describe an efficient method of detecting this species in the field using eDNA, and (iii) identify species that remain targets for sequencing from herbarium collections. While we do not resolve the species’ origin, the tools and approach described here establish a transferable framework for tracing cryptogenic algal introductions, especially in remote or under-sampled regions where biogeographic approaches are constrained by limited baselines and sparse records.

Materials and Methods

Oceanographic modeling

Oceanographic information was acquired from the Hybrid Coordinate Ocean Model (HYCOM) with the Navy Coupled Ocean Data Assimilation 1/12° Global Ocean Forecasting System 3.1: 41-layer Reanalysis (GLBv0.08 Experiment 53.X daily snapshot) (Cummings, 2005; Cummings & Smedstad, 2013). The dataset consisted of surface currents from January 1, 2000 to December 30, 2015 between the coordinates 90°E–55°W and 80°S–90°N, exceeding the bounds of the entirety of the Pacific Ocean. This time frame includes over a decade of oceanographic conditions leading up to the earliest satellite observation of C. tumulosa at Manawai in 2015 (Lopes et al., 2023).

HYCOM oceanographic information was passed to the individual-based biophysical stochastic lagrangian dispersal simulator, the Connectivity Modeling System (CMS) (Paris et al., 2013). CMS can run a backtracking module that traces a particle found at a particular place and time backwards to its estimated source location (Paris et al., 2013). Model runs released 100 particles in daily releases from Manawai (27.8333°N, 175.8333°W) for each day from January 1, 2000 to December 30, 2015. While C. tumulosa is present at all three of the northernmost atolls of PMNM, Manawai was selected as the modeled release location because it has the highest recorded abundance and biomass of the alga among the atolls of PMNM (Nichols et al., 2025) and was the earliest documented location of its detection (Sherwood et al., 2020). Each particle was restricted to the surface ocean (flag upperlevelsurface = true) and resolved at daily intervals. Surface transport of lightweight polymers accounts for the majority of marine debris in Hawai‘i (Brignac et al., 2019). Restriction to the surface ocean is further supported by modeling of C. tumulosa dispersal in PMNM, which indicated that higher density objects have significantly lower dispersal potential—even between neighboring atolls (Fumo et al., 2024). Potential release areas were created by extracting 1° hexagonal grids intersecting with the Global Territorial Sea 12 nm shapefile from the Pacific Data Hub (Flanders Marine Institute, 2023; QGIS, 2024). Hexagonal grid settlement regions better capture the contours of coastlines and are more suitable than rectangular grids for individual based modeling (Birch, Oom & Beecham, 2007). Hexagonal settlement regions encompassing the Hawaiian archipelago were removed from the shapefile prior to inclusion in the CMS runs to prevent the possibility of self-recruitment back to the region of origin where the species is unlikely to have originated (Sherwood et al., 2020; Carlton & Schwindt, 2024; Nichols et al., 2025). Further, since one of the objectives of this study was to narrow the scope of potential source locations for C. tumulosa, excluding self-recruitment into Hawai‘i—where surveys are already ongoing—maximized the utility of the results. All particles were given a horizontal diffusivity of 10 m2/s (Okubo, 1971; Fig. S1).

Data downloads and model runs were conducted using the CMS functions getdata and cms, respectively on the University of Hawai‘i’s High Performance Computing (HPC) cluster (https://datascience.hawaii.edu/hpc). The number of particles arriving in each hexagon (landings) was mapped and higher-resolution mapping of each region displaying increased likelihood of settlement was conducted using the sf (Pebesma, 2018; Pebesma & Bivand, 2023) and mapdata (Becker, Wilks & Brownrigg, 2022) libraries in R (R Core Team, 2023). Regions of interest were established surrounding all high-likelihood settlement areas indicated in the model from the backward trajectory start point, Manawai (Fig. S2). These regions of interest were: Japan’s Eastern coast from the Izu Islands to Hokkaidō (29–43.5°N; 135–150°E), The central Pacific encompassing Johnston Atoll and the Line Islands including Palmyra (0.5–18°N; 155–172°W), and the Eastern Tropical Pacific islands of Clipperton and the Galápagos Archipelago (2.5°S–12°N; 88.5–110°W). The drift time to each region of interest was assessed through pairwise comparisons using Wilcoxon rank sum tests with Bonferroni correction (R Core Team, 2023). The timing of arrivals to Manawai, categorized by regions of interest, was analyzed in relation to the phase of the El Niño Southern Oscillation (ENSO) and by season of the year. Seasons were defined as September-October-November, December-January-February, March-April-May, and June-July-August. These analyses were plotted and tested for significance using a Wilcoxon paired rank sum test with Bonferroni correction (R Core Team, 2023). ENSO phase conditions in each month over the modeled period were downloaded using the download_enso function in the rsoi library (Albers, 2023) in R.

Diffusivity value optimization was conducted through comparison of targeted CMS runs with real-world drifter data. Marine debris at Manawai was tagged using Satlink solar-powered satellite buoys weighing 13.7 kg with a 40.6 cm diameter and 36.8 cm height in September-October 2018. Units were set to record their position once every 4 h. Of the six tracking devices attached to marine debris, two loggers became detached and escaped the atoll on February 1, 2020 and December 14, 2020, respectively. Both loggers stopped recording in September 2021 and drifted a considerable distance from Manawai. For comparison with CMS output, we utilized HYCOM data from February 1, 2020 to September 30, 2021 for surface currents between the coordinates 150°E–160°W and 15–40°N. CMS runs were conducted in backtracking mode independently for each logger with 1,000 releases from the final GPS coordinate and timestamp of each receiver. Each release was replicated five times with each iteration varying only in horizontal diffusivity which was set to 0.5, 5, 10, 20, or 50 m2/s. The optimum diffusivity value for drifting nets in the backtracking model was assessed by calculating the minimum distance that each particle drifted from Manawai, the true release location of the loggers, using the geosphere library (Hijmans, 2022) in R. The performance of each turbulence value was evaluated through a pairwise Wilcoxon rank sum test with Bonferroni correction implemented in R through the pairwise.wilcox.test function with the argument p.adjust.method set to ‘bonferroni’ (R Core Team, 2023) (Fig. S1).

eDNA collection and screening

Three potential source regions identified by oceanographic modeling were sampled for C. tumulosa eDNA during 2024: Kiritimati Atoll of the Line Islands (11), Johnston Atoll (20), and Japan’s Okinawa Island (10) (Fig. S3; Table S1). Water samples for eDNA analysis were collected under USFWS Special Use Permit 12543-24001 (Johnston Atoll), a Research Consent Certificate from the Ministry of Fisheries and Marine Resources Development (Kiribati), and with the permission of the University of the Ryukyus (Okinawa). At each site within a region, duplicate 2-L seawater samples were collected from within 1 m of the bottom and 2–3 m from each other to maximize the probability of detecting C. tumulosa in low abundance (Nichols et al., 2025). Each 2-L biological sample and a tap water control (see contamination prevention, below) were shaken, filtered through mixed cellulose ester filters (Millipore; diameter: 47 mm; pore size: 0.22 µm) on a peristaltic pump (Cole-Parmer, Vernon Hills, IL, USA), and frozen in liquid nitrogen. DNA was extracted (DNeasy Blood & Tissue kit, Qiagen, USA) from thawed filters which were cut in half as described in Nichols & Marko (2019) and Nichols, Timmers & Marko (2022).

To increase screening efficiency, eDNA from individual replicate water samples were pooled within each region and amplified using a qPCR assay for C. tumulosa (Nichols et al., 2025). To ensure that pooling multiple samples during the screening process did not create false-negative detections (due to dilution of presumably low concentrations of C. tumulosa eDNA), a second pool was also created per region, with an addition of 1 µL eDNA from a field site at Kuaihelani, PMNM, where C. tumulosa was visually observed in 1% relative abundance (hereafter referred to as the “field positive”). Conceptually, pooled regions with spiked positive eDNA (hereafter referred to as the “positive pool”) should produce amplifications using the assay, indicating the presence of C. tumulosa within the pooled region. Pooled regions without the spiked eDNA (hereafter referred to as the “sample pool”) should only amplify if C. tumulosa is locally present.

Reactions were run using triplicate technical replicates consisting of 4.5 µL SYBR green SSo Advanced Supermix (Bio-Rad, Hercules, CA, USA), 3 µL ultrapure H2O (Growcells), 0.5 µL bovine serum albumin (20 mg/mL; Thermo Fisher Scientific), 0.5 µL (10 μM) of each of the forward (5′-GCCGTGAATCGTTCTATTGC-3′) and reverse (5′-TCAGCTCTTTCGTACATATTCTCC-3′) primers (Nichols et al., 2025) that were designed to amplify a 95 bp fragment of rbcL specific to C. tumulosa, and 1 µL sample pool eDNA. Amplifications were run on a CFX96 Touch Real-Time PCR Detection System and CFX Manager software (Bio-Rad, Hercules, CA, USA). The CFX96 calculated critical thresholds of fluorescence (above which detections can be discriminated from background noise), averaged across amplification plates. A positive DNA starting quantity (calculated automatically in CFX Manager against a tenfold serial dilution of standards with known DNA concentrations) in any technical replicate was considered a positive detection. Amplification curves were plotted using ggplot2 (Wickham et al., 2016) in R (R Core Team, 2023) and fitted with generalized additive model (GAM) smoothers. Positive detections were considered reliable if the GAM smoother exceeded the mean threshold of fluorescence, demonstrating precision among replicates.

All laboratory surfaces and equipment were routinely decontaminated using a 10% bleach solution. Water collection containers were sterilized with 10% bleach for 12 h, air dried, and rinsed with surface seawater at each new location prior to sampling. Equipment contamination was monitored using bleach-sterilized filtration equipment and a 1-L tap water or DI water equipment blank (EB) filtered prior to field samples, at least once per day, which served as a negative control for both the filtration and DNA extraction steps. PCR contamination was monitored with triplicate PCR negatives (no-template controls, NTC) per 96-well plate (Nichols et al., 2025).

DNA barcoding of herbarium specimens

Sequencing was attempted from 156 preserved specimens including 60 pressed algal specimens from the Herbarium Pacificum of the Bernice P. Bishop Museum in Honolulu, Hawai‘i, USA (BISH), 39 from the herbarium of Meise Botanic Garden (BR) (formally from Ghent University, GENT), 26 from the Environmental Research Institute, São Paulo, Brazil (SP), 14 from the National Institute of Water and Atmospheric Research, Wellington, New Zealand (NIWA), seven from the University of Malaya in Kuala Lumpur, Malaysia (KLU), four from the Prince of Songkla University, Hat Yai, Thailand (PSU), and six from C. tumulosa collected in PMNM (Table S2). New Zealand specimens were collected under NIWA (National Institute of Water and Atmospheric Research) Biodiversity, current project OCBR2401.

Genomic DNA was extracted following a modified cetyltrimethylammonium bromide (CTAB) protocol (Doyle & Doyle, 1989) with a CTAB buffer and β-mercaptoethanol as a grinding solution as described in Fumo & Sherwood (2023). The Universal Plastid Amplicon (UPA) region (Presting, 2006) was amplified following Sherwood & Presting (2007) with the primer pair p23SrV_f1 (5′-GGACAGAAAGACCCTATGAA-3′) and p23SrV_r1 (5′-TCAGCCTGTTATCCCTAGAG-3′). Cycling conditions followed those of Sherwood & Presting (2007) with successful PCR amplifications confirmed using gel electrophoresis and purified with ExoSAP-IT (Affymetrix, Santa Clara, California, United States) before submission to GENEWIZ (Azenta Corporation, South Plainfield, NJ, USA) for Sanger sequencing. Sequences generated in this study were uploaded to GenBank and assigned accession numbers PV036782–PV036852 (Table S2).

Forward and reverse reads were assembled in Geneious Prime 2024.0.3 (https://www.geneious.com) and aligned with a reference database. This database was constructed by retrieving other closely related sequences using an existing C. tumulosa UPA sequence (NC_057618), which was compared using the National Center for Biotechnology Information (NCBI) Basic Local Alignment Search Tool (BLAST) to recover the 100 nearest matches from GenBank. While obvious contaminants were removed, no further taxonomic reassignment was performed beyond the GenBank name or the original collector’s identification. The goal of this analysis was not to identify each sequence as a species but rather to determine whether any sequences matched C. tumulosa. Additional UPA sequences for C. tumulosa were generated using the extracted DNA generated during the description of the species (Sherwood et al., 2020).

All Rhodomelacean sequences generated from the BLAST search (n = 84) were aligned with those from the successful amplifications of C. tumulosa from PMNM (n = 6) and those generated from material stored in herbaria (n = 66) using MUSCLE v5.1 in Geneious (Edgar, 2022) to create a final alignment of 159 UPA sequences of 389 bp in length (Supplemental Information). Percent identity was recorded for each sequence in comparison to C. tumulosa, NCBI accession number NC_057618 (Paiano et al., 2021) and organized according to similarity (Table S2). Additional BLAST and Barcoding of Life Database (BOLD) Systems searches of the C. tumulosa SSU (BLAST only), COI, and rbcL sequences generated by Sherwood et al. (2020) were conducted to confirm that no matching sequences had been uploaded since the initial description of the species.

Geographic distributions of Chondria species

All entities classified under the genus Chondria C. Agardh were investigated for range overlap with the regions of interest outlined by model results. Entities that had been synonymized with taxa in other genera were included in this analysis unless the nomenclatural change removed them from the family Rhodomelaceae altogether. Detailed distribution information was obtained from AlgaeBase (Guiry & Guiry, 2024). For each entity the GenBank NCBI and BOLD Systems databases were queried and the DNA barcodes available for each were recorded (Table S3).

Results

Oceanographic modeling

Landing totals for 584,300 particles tracked backwards in time (Table S4) indicated that the most likely regions of origin for C. tumulosa in PMNM suggested by CMS were (i) Japan (27.4% of released particles), (ii) the central Pacific islands of Johnston Atoll and the Line Islands including Palmyra Atoll (23.8%), and (iii) the ETP islands of Clipperton and the Galápagos archipelago (14.3%) (Fig. 1). Here, "landings" refers to the endpoints of backtracked particles—the modeled location of a particle’s origin. Additional landings occurred throughout the regions outside of these boundaries in the central, western, and eastern Pacific at lower frequency. Landings in individual hexagons were generally low with 18 of the 3,782 modeled polygons supplying ≥1% of particles to Manawai (Fig. S2).

Figure 1 Total modeled landings at source locations for Chondria tumulosa backtracked from Manawai (Pearl and Hermes Atoll), Hawai‘i.

From December 30, 2015 to January 1, 2000, 100 particles per day (584,000 total) originating at the backward trajectory start point, Manawai, were traced until landing in a potential source location. Hexagonal polygons were the regions designated as landing areas in the Connectivity Modeling System. Each hexagon is colored by the number of landings in that settlement polygon throughout the model duration with colorless hexagons receiving zero landings throughout the modeled period. All landings throughout the modeled region are shown in (A). The regions of highest landings were Japan (B), the central Pacific islands of Johnston Atoll, and the Line Islands including Palmyra (C), and the eastern Pacific islands of Clipperton and the Galápagos archipelago (D). The legend in (A), indicating landings, applies to all panels. Manawai is indicated by a gray triangle, the Main Hawaiian Islands (MHI) and Papahānaumokuākea Marine National Monument (PMNM) are indicated.

The six individual landing polygons with the highest likelihood of supplying C. tumulosa to Manawai were geographically concentrated in four locations. Two polygons were located in Japan: Cape Inubo with 44,336 particles (7.6%), and the Izu islands to the south of Tokyo with 24,415 (4.2%). Two polygons were at Johnston Atoll: the western polygon with 43,248 (7.4% of released particles) and the northeastern polygon with 28,405 (4.9%). The remaining two polygons were in the Galápagos archipelago: the northwestern corner, centered around the islands of Darwin and Wolf, with 32,133 (5.5%), and the second most northwestern polygon between Isabela and Wolf Islands with 23,149 (4.0%).

Modeled landings were observed along Japan’s eastern coastline with a peak in origin likelihood (landings) occurring at Cape Inubo (Fig. 1). Johnston Atoll is relatively small in comparison to the surrounding hexagonal grids with high landings across the entire area and an apparent peak in landings on the western edge of the region. The Line Islands including Palmyra Atoll exhibited lower landings overall with a local peak in the northern ends of the islands and island chain (Fig. 1). The Galápagos Islands in the ETP showed a peak in landings at the northwestern end of the archipelago in the grids surrounding the islands of Darwin and Wolf while Clipperton Island showed relatively low likelihood throughout the region (Fig. 1).

Particles landing in Japan were generally tracked on a northerly route from Japan through the North Pacific Subtropical Gyre (i.e., Kuroshio, North Pacific, California, and North Equatorial Currents) to Manawai, while particles landing in the ETP largely traveled through the central Pacific via the North Equatorial Current before landing at Manawai. Central Pacific particles generally traveled to the north and west, taking a more direct path towards Manawai. The combined locations of all tracks centers around Manawai, tapering to the west towards Japan and towards the south and east towards the ETP with an extension towards the southwest and nearer Micronesia (Fig. 2; Fig. S4).

Figure 2 Density cloud of particle locations throughout the modeled backtracking period from Manawai (Pearl and Hermes) Atoll, Hawai‘i.

Colors indicate the proportion of particles located in each pixel throughout the study period, January 1, 2000 to December 30, 2015 using the Connectivity Modeling System. Particles originating at the backward trajectory start point, Manawai (indicated by a gray triangle), generally tracked towards the northwest or southeast until landing in a potential source region. Warmer colors represent regions with a higher concentration of particles, while cooler colors indicate areas with fewer particles during the modeled duration. The high concentration of particles surrounding Manawai reflects the start point of the model run. Currents influencing the dispersal of particles are shown in gray arrows and named. The North Pacific Subtropical Gyre is composed of the Kuroshio Current, North Pacific Current, California Current, and North Equatorial Current, which runs north of the eastward flowing Equatorial Counter Current. The Hawai‘i Lee Counter Current (HLCC) and Subtropical Counter Current (STCC) occur within the gyre, moving more weakly from west to east and are represented by thinner arrows.

The mean drift time for landing in each region from the backward trajectory start point differed significantly at 420, 552, and 1,007 days for the central Pacific, Japan, and the ETP, respectively (Wilcoxon p = 2.2e−16 in all cases) (Fig. S5). The number of particles landing at their respective origin locations per month varied considerably through time, yet the number of landings in the ETP and Line Islands including Palmyra did not vary with ENSO phase (i.e., Neutral, El Niño, or La Niña) at the time of landing (Wilcoxon p > 0.24 and shown in Table S5). Neutral period landings—those outside of both El Niño and La Niña periods—from Japan were significantly higher in comparison to both El Niño and La Niña landings (p = 0.038; p = 0.003, respectively). There were no significant differences in landings by season from any of the source regions of interest (Wilcoxon p > 0.41 in all cases and shown in Table S6).

The modeled minimum passing distance of particles released from the final correspondence point of marine debris tagged at Manawai were compared between trials where diffusivity values were equal to 0.5, 5, 10, 20, and 50 m2/s. Mean (median) minimum passing distances for each diffusivity value were 160.7 (136.2), 133.5 (119.3), 125.9 (113.7), 129.2 (113.6), and 140.6 (117.1) km, respectively and the central tendencies of the groups varied significantly (Kruskal-Wallis p = 2.2e−16). A Wilcoxon signed rank test indicated that the minimum passing distance in group 0.5 m2/s differed from all other groups (Table S7). Groups 5, 10, 20, and 50 m2/s did not differ (Fig. S1; Table S7).

Genetic analyses of eDNA and herbarium specimens

Following Darling, Jerde & Sepulveda (2021), we use the term ‘presumed negative’ to describe regions where eDNA indicates absence, but this has not been confirmed by non-eDNA observations. For eDNA screening at Kiritimati, Line Islands, Johnston Atoll, and Okinawa Island, Japan (Table S1), all the sample pools were presumed negative, lacking any discernible traces of C. tumulosa eDNA (Fig. 3A). Target eDNA was detected in all positive pools, as expected, with concentrations below 10 copies per reaction (Fig. 3B), indicating that the pooling of samples does not mask the detection of a single positive site.

Figure 3 Screening of pooled eDNA samples from potential source regions.

(A) Amplification curves (in relative fluorescence units, RFU) and (B) eDNA concentrations (“starting quantity” ± SE) using the species-specific qPCR assay for Chondria tumulosa against standards with known copy numbers. Samples from Kiritimati, Line Islands (n = 22), Johnston Atoll (n = 40), and Okinawa, Japan (n = 20) were pooled by region (“sample pool”) and amplified in triplicate. A second set of pooled samples per region (“positive pool”) were each spiked with eDNA from a confirmed positive site (“field positive”, n = 2) in Papahānaumokuākea National Marine Sanctuary, where C. tumulosa was observed at 1% relative abundance. DNA extracted directly from a preserved C. tumulosa specimen (“tissue”, n = 1) and no-template controls (“NTC”, n = 4) are also shown. Critical thresholds of fluorescence (above which detections can be discriminated from background noise) are plotted with a dashed line.

Specimens bearing a superficial morphological resemblance to C. tumulosa were sampled from preserved pressed specimens from different herbaria (n = 156). Attempts to amplify the UPA marker were successful for 65 of these (Table S2) and none of the resulting sequences returned BLAST matches to C. tumulosa. The nearest match to C. tumulosa based on the UPA marker was accession number ARS 11026 (BISH 682104) with a 97.5% similarity. This specimen, labeled Chondrophycus cartilagineus (Yamada) Garbary & J.T. Harper, was collected from Pohnpei, Federated States of Micronesia in 1996 (Table S2). This similarity exceeds the nearest match available on NCBI GenBank (HQ421169-Chondria dangeardii E.Y. Dawson from Hawai‘i) which was a 95.8% match. BLAST and BOLD Systems similarity searches of the sequences generated by Sherwood et al. (2020) were conducted with rbcL. COI, and SSU (BLAST only) and did not reveal any matches or near-matches (>99%) to C. tumulosa with highest pairwise identities of 89.8%, 94.5%, and 97.2%, respectively. These results indicate that C. tumulosa is not currently represented in the BLAST or BOLD Systems databases and our search through global herbaria did not identify a specimen matching the species.

In total, 113 species of Chondria are either currently accepted taxonomically, of unresolved taxonomic status, or have been transferred to other genera within the family Rhodomelaceae (Guiry & Guiry, 2024). The genus currently includes 78 species that are currently accepted, while 69 have been placed in synonymy. Our comparison with geographic species ranges, based on targeted searches through AlgaeBase (Guiry & Guiry, 2024), found that of the 113, 25 species exhibit range overlap with potential source regions identified through the modeling analysis—18 in Japan, three in the Line Islands including Palmyra, two in Johnston, two in Clipperton, and two in the Galápagos Islands (Table S2; Table S3). Additional searches through NCBI GenBank and BOLD Systems databases indicated that DNA barcodes are not available for 12 of these 25 species.

Discussion

Uncovering the source of cryptogenic species is a globally relevant challenge in invasion biology and biogeography, particularly in marine systems where long-distance dispersal, limited historical baselines, and taxonomic uncertainty can obscure species origins (Carlton & Schwindt, 2024). Lack of information on the native range of an organism limits our understanding of the ecological and evolutionary background of the species, complicating the prediction of its impacts and any evolutionary shifts that occurred during the introduction (Hierro, Maron & Callaway, 2005; Geller, Darling & Carlton, 2010; Buckley & Catford, 2016). For instance, if C. tumulosa is indeed an introduced species in Hawai‘i, there may be herbivores of C. tumulosa in its native range that do not exist in PMNM which could serve as biological control agents. Further, population genetic studies can help uncover patterns of reproductive system variation and gene flow (e.g., see work in G. vermiculophylla, Krueger-Hadfield et al., 2016, 2017). Oceanographic modeling indicates the most likely sources for the arrival of C. tumulosa in PMNM are Japan, the central Pacific, encompassing Johnston and the Line Islands including Palmyra Atoll, or the ETP islands of Clipperton and the Galápagos Archipelago. Despite signs that the species may be introduced into the region (Sherwood et al., 2020; Lopes et al., 2023; Fumo et al., 2024; Williams et al., 2024), its introduction pathway remains unknown. The establishment of a narrowed set of search regions for environmental detections increases the likelihood of identifying the C. tumulosa source which would allow for essential research into the origin, evolutionary history, ecological traits, and potential for control of C. tumulosa, aiding in safeguarding the protected and culturally significant PMNM (Wiener & Wagner, 2013; Gaymer et al., 2014; Kikiloi et al., 2017).

Although PMNM is not as strongly influenced as the MHI by anthropogenic factors such as overfishing, eutrophication, and sedimentation, PMNM still faces global challenges (Selkoe et al., 2009). Climate change impacts, such as ocean warming (Sarmiento et al., 2004), contribute to marine heat waves (Frölicher, Fischer & Gruber, 2018) leading to mass coral bleaching (Couch et al., 2017). Additionally, climate change may cause shifts in hurricane intensity and storm paths (Mendelsohn et al., 2012; Wehner, Zarzycki & Patricola, 2019). These events not only cause widespread damage (Pascoe et al., 2021) but also have the potential to further distribute marine debris (e.g., Hu et al., 2023) and non-native species (e.g., Steiner et al., 2010), including C. tumulosa (Lopes et al., 2023). Other global challenges facing PMNM are sea level rise impacts (Shope, Storlazzi & Hoeke, 2017), as well as the accumulation of marine debris (Royer et al., 2023; Benadon et al., 2024) which is increasingly recognized as a vector for non-native species introductions (Carlton et al., 2018; Chong et al., 2023; Haram et al., 2023; Carlton & Schwindt, 2024). Further, C. tumulosa continues to spread in PMNM (Lopes et al., 2023), likely aided by marine debris transport (Fumo et al., 2024) though other potential vectors of introduction also exist and include ballast water and hull fouling of recreational and commercial vessels. The introduction of the species to the MHI is a major concern because the archipelago has a history of economically and ecologically taxing marine algal invasions (Smith, Hunter & Smith, 2002; Conklin & Smith, 2005; Veazey et al., 2019), including further spread aided by asexual reproduction (Thornton et al., 2024).

The oceanographic modeling explicitly targeted marine debris as a dispersal vector across the expanse of the Pacific Ocean and included the incorporation of real-world GPS information from tagged flotsam to optimize the model diffusivity parameter. The passing distance (i.e., their minimum distance to Manawai) of modeled marine debris backtracked from their open ocean endpoint was not significantly affected by diffusivity values between 5–50 m2/s. This indicates that CMS modeling using global HYCOM data and constrained to the surface ocean may be insensitive to the selection of a diffusivity value within this range. In this case, the shortest passing distance was observed in the 10 m2/s model, which we consider a reasonable default where an informed estimate is lacking. Chondria tumulosa may be suited to passive rafting via entanglement, as it has been observed drifting while attached to flotsam or as detached fragments carried by currents (Lopes et al., 2023). Upon arrival in a new site or atoll, the ability to fragment and grow likely facilitates rapid spread and accumulation—as evidenced by the genetic signatures of asexual reproduction described at Manawai (Williams et al., 2024). Rafting may not only transport the species within the northern PMNM, but also may be responsible for its initial introduction (Fumo et al., 2024). Although Hawai‘i is one of the most isolated archipelagos in the world, there are many routes for C. tumulosa to enter the region. A factor that has generated increasing concern in recent decades is that Hawai‘i in general, and PMNM in particular, are near the Great Pacific Garbage Patch. This is perhaps the world’s largest collection of drifting flotsam, including debris from across the Pacific (Maximenko, Hafner & Niiler, 2012), and hosts living communities of coastal species in pelagic environments (Haram et al., 2023). Particles landing at Manawai in the CMS output were derived from numerous locations across the expanse of the Pacific Ocean, demonstrating this reality. Nevertheless, our model results narrow the potential drifting routes and for particles coming from their respective source regions and thus aid in the broader search for C. tumulosa.

Rafting particles transiting from the ETP largely pass through the central Pacific and are subsequently connected with Hawai‘i via the North Equatorial Current, Hawai‘i Lee Counter Current, North Hawai‘i Ridge Current, and Subtropical Counter Current (Maximenko, Hafner & Niiler, 2012). Evidence of this pathway’s potential to transport C. tumulosa into the central Pacific comes from pumice originating from the eastern Pacific’s Revillagigedo archipelago being found in the Line Islands (Kiritimati Island) and also in Hawai‘i, though in lower abundance than at Kiritimati (Jokiel & Cox, 2003). Many of the particles backtracked to the ETP from Manawai generally passed through the central Pacific during their transit, narrowly missing the Line Islands and neighboring archipelagos. Thus, particles departing the ETP may settle in the central Pacific prior to their transit northward into Hawai‘i.

Alternatively, connectivity of PMNM with Japan and Indo-Polynesia is well-documented, because the species makeup of macroalgae in PMNM consists mainly of a mix of species with affinities to these regions (McDermid & Abbott, 2006; Randall, 2007; Kawai et al., 2023; Kittle et al., 2024). Algal connectivity with Japan was further evidenced during the earthquake of 2011 in Tōhoku—a region of Japan with high landings in the model output—which generated marine debris that transited across the North Pacific via the Subtropical Gyre into Hawai‘i (Carlton et al., 2017). The Tōhoku earthquake transported numerous algae out of Japan including a previously undescribed species recovered from drift material (West et al., 2016), species which exhibited close phylogenetic relationships with those of Tōhoku (Hanyuda, Hansen & Kawai, 2018), and many with high invasion potential (Hansen, Hanyuda & Kawai, 2018).

The Micronesian region showed a lower level of landings in comparison to the Central Pacific and Japan. We note however that particles originating from Indo-Polynesia, including Micronesia, typically enter Hawai‘i through the use of intermediary islands including, among others, Johnston Atoll, Kingman Reef, and the Line Islands (Kobayashi, 2006; Skillings, Bird & Toonen, 2011). Johnston Atoll in the central Pacific is often considered an outpost of PMNM (Skillings, Bird & Toonen, 2011) being primarily connected to the central islands near Lalo (French Frigate Shoals) (Grigg, 1981; Kobayashi, 2006). However, if Johnston Atoll is considered part of the Hawaiian biogeographic region (Crandall et al., 2019), then if the species were to be identifyied there, it would not fully resolve the species origins outside of Hawai‘i.

With a lack of C. tumulosa eDNA detections from Johnston Atoll and from Kiritimati Atoll in the Line Islands, should the species have arrived at Manawai from Micronesia, it may have done so by transiting via the North Equatorial Counter Current, Subtropical Counter Current, or North Pacific Subtropical Gyre and circumventing these islands (Hu et al., 2015; Wang & Wu, 2018). The North Equatorial Counter Current flows at very low latitudes, making it likely that modeled particles following this route would encounter multiple archipelagos en route to Hawai‘i, thereby reducing the modeled likelihood of successful transit without contacting another settlement polygon, and thus decreasing the modeled observed landings there. While the modeled likelihood of transiting through this region is lowered due to the presence of islands, those islands may act as stepping stones which increase the likelihood of successful transport. Thus, Micronesia and the intermediary islands en route to Hawai‘i remain targets for eDNA searches. The Subtropical Counter Current, which runs in an easterly direction at roughly the latitude of the Micronesian region, strengthened in the decades preceding the discovery of C. tumulosa in the PMNM (Wang & Wu, 2018) suggesting an increased potential to transport marine debris.

Alongside these dynamics are the ENSO-derived variations in the currents of the North Pacific (Hu et al., 2015). Despite the strong role of ENSO in the ETP (Wang & Fiedler, 2006), the phenomenon did not impact the likelihood of particle landings from this region nor did it impact the central Pacific landings. Landings from Japan were highest during neutral phase ENSO periods. This pattern is likely associated with the changes in circulation and marine debris concentration patterns in the North Pacific associated with ENSO (Howell et al., 2012). The incidence of marine debris in PMNM increases (decreases) during El Niño (La Niña) periods because of the southward (northward) migration of the Subtropical Convergence Zone (STCZ); however, the arrival of debris remains high regardless of ENSO phase (Morishige et al., 2007). During El Niño (La Niña), though, the Kuroshio Current weakens (strengthens) and may impact the exportation and retention of marine debris in the associated recirculation gyre (Howell et al., 2012). Elevated neutral period landings from Japan to Manawai are thus likely associated with the change in strength of the Kuroshio and movement of the STCZ to a favorable median condition.

Clearly, much variability exists in the pathways and source regions in the CMS output. The model primarily captured the drift routes and source regions with the highest probability of providing drift material to Manawai despite the existence of many other lesser highlighted routes. Though there is structuring of marine populations across the Indo-Pacific (e.g., Keith et al., 2013; Bowen et al., 2016), the region is generally well-connected (Crandall et al., 2019). Hence, a confirmed detection of C. tumulosa in the Indo-Pacific may suggest the species has a broader distribution and is not endemic to the detection site. Indeed, there is also the possibility that C. tumulosa has been exported from PMNM to other regions. Basin-scale searches for C. tumulosa will remain warranted even after initial detections outside of PMNM and should be followed by genetic analyses to determine structure of the population throughout its native range in order to identify the pathway of introduction to PMNM and provide a more complete picture of the species’ introduction history.

The varying mean modeled rafting durations of C. tumulosa to Manawai are long, at 421 days from Japan, 587 days from the central Pacific, and 1,014 days from the eastern Pacific. Generally, organisms best suited to rafting are those which do not feed on their host raft, reproduce asexually, are small, able to hold onto floating objects, establish and compete successfully on rafts, and persist during their voyage (Thiel & Gutow, 2005). There is evidence that C. tumulosa exhibits these traits because it has been observed rafting in PMNM (Lopes et al., 2023), can persist via thallus fragmentation (Williams et al., 2024), may occur cryptically (Nichols et al., 2025), and can dominate otherwise healthy reef systems (Sherwood et al., 2020). Further, for C. tumulosa to raft effectively, it must reattach to the substrate, a trait it likely possesses because it attaches via haptera (Sherwood et al., 2020). Thus, spore production may not be necessary for initial establishment, and the dominance of the tetrasporophyte phase (Williams et al., 2024) suggests the life cycle could be reestablished in its new environment (Krueger-Hadfield, 2020). Debris from the 2011 Tōhoku earthquake began landing in North America and Hawai‘i ~300 days after the earthquake (Carlton et al., 2017), indicating that transit times across these great distances can be rapid. Survival over long periods was also observed following this event, as a large mass of buoys and ropes derived from Tōhoku arrived in Hawai‘i >2,000 days after the earthquake with living biota aboard, including algae (Carlton et al., 2017, 2018). Landings in Japan occur more rapidly and regularly than those passing through the formidable Eastern Pacific Barrier where corals, among other organisms, are generally isolated from their central and western Pacific counterparts (Baums et al., 2012). However, modeling of coral propagule dispersal assuming a 120-day competency period suggests corals successfully settle from the ETP to the central Pacific on a regular basis (Connolly & Baird, 2010; Wood et al., 2016). Further, global modeling assessing the transit times of particles into the world’s oceanic accumulation centers in subtropical gyres suggests that particles arriving into the North Pacific garbage patch have a minimum transit time from Japan, the ETP, and the central Pacific of <250 days (Maximenko, Hafner & Niiler, 2012). The broad scale movement of living biota attached to marine debris across oceanic basins in short time spans can and does occur episodically. The drift times from the CMS run in this study vary greatly, suggesting the model captured a wide range of potential drifting pathways and potential lifespans with some relatively short transits occurring from all regions of interest falling within the realm of possibility, especially considering C. tumulosa may have been attached to marine debris and is likely to disperse via thallus fragments rather than spores that have a much shorter longevity.

Though our search did not result in any detections of C. tumulosa from herbarium collections, we acknowledge that future methodological advances may improve recovery of DNA from archival material. Further, re-collections of archival material from original collections sites can aid in resolving the taxonomy of this group. While morphological identification of Chondria is challenging, the characteristics outlined by Sherwood et al. (2020) provide useful guidance for targeting specimens that may warrant DNA barcoding. Of the currently accepted Chondria species (Guiry & Guiry, 2024) with overlapping distributions with the modeled regions of interest, C. acrorhizophora Setchell & N.L. Gardner, C. clarionensis Setchell & N.L. Gardner, C. econstricta M. Tani & Masuda, C. flexicaulis W.R. Taylor, C. lancifolia Okamura, C. minutula Weber Bosse, C. repens Børgesen, C. simpliciuscula Weber Bosse, C. stolonifera Okamura, and C. xishaensis J.-F. Zhang & B.-M. Xia have no verified DNA sequences and should be targeted from global herbaria as a meaningful contribution in the search for matches to C. tumulosa (Holmes, 1896; Setchell & Gardner, 1930; Taylor, 1945; Dawson, 1963; Buggeln & Tsuda, 1969; Segawa, 1981; Yoshida, Nakajima & Nakata, 1990; Yoshida, 1998; Tani & Masuda, 2003; Titlyanov et al., 2006; Serviere-Zaragoza et al., 2007; Ruiz & Ziemmeck, 2011; Tsuda, Fisher & Vroom, 2012; Titlyanov & Titlyanova, 2012; Tsuda & Walsh, 2013; Yoshida, Suzuki & Yoshinaga, 2015; Sutti et al., 2018; Titlyanov et al., 2019). A phylogenetic reassessment of the genus Chondria and the tribe Chondrieae is warranted to clarify the systematics of this diverse group and enhance our understanding of the distribution of C. tumulosa. This framework can be extended to other cryptogenic red algae of unknown origin such as the putative Japanese Pyropia J. Agardh which appeared on the central coast of British Columbia, Canada, in 2015 (Lindstrom, 2018).

Our recommended procedures for investigating model-guided source locations in the search for C. tumulosa are as follows: searches should continue by screening eDNA collected from regions of interest (e.g., Japan, Galápagos, Line Islands including Palmyra, Johnston Atolls) because this method increases sensitivity and minimizes survey costs (Jerde et al., 2011; Dejean et al., 2012; Smart et al., 2015; Nichols et al., 2025). Our screening of eDNA sample pools from the model-guided potential source regions did not detect C. tumulosa. Although false negatives cannot be entirely ruled out, using duplicate water samples and triplicate qPCR replicates per site has been shown to substantially reduce the likelihood of both false-positive and false-negative detections (Nichols et al., 2025). Nevertheless, false negatives may still occur at a regional scale due to logistical constraints in sampling all reef habitats across the three regions. Therefore, further investigation into these regions may be warranted. The power of eDNA analysis lies in the ability to search for C. tumulosa in leveraged samples that were designed to study other organisms. In this manner, the search for C. tumulosa using eDNA can be optimized by opportunistically checking for detections using pooled samples from any collection region of interest, provided sample pools do not dilute target eDNA any lower than the present study; we suggest limiting pooled samples to <20 for reliable amplification of potential low-abundance sites. A positive eDNA detection of C. tumulosa may then be considered as a strong rationale to individually re-screen sites and initiate in-water surveys. The species can exhibit robust, cryptic, or epiphytic growth patterns, necessitating that phycologically trained in-water divers meticulously examine potential habitats and cryptic microenvironments. Specimens collected during in-water surveys should be cleaned of epiphytes and debris and stored in silica desiccant when in remote field sites or, if possible, flash frozen in liquid nitrogen. The remainder of the collected specimen should be examined for morphological similarity to C. tumulosa via microscopy and pressed to make an herbarium voucher. Samples matching C. tumulosa morphologically—or nearly so—should be extracted and sequenced for the UPA barcode (Sherwood & Presting, 2007) for comparison to published UPA sequences of C. tumulosa (Sherwood et al., 2020; Paiano et al., 2021). Any matches of high similarity should be sequenced additionally for the plastidial rbcL, mitochondrial COI, and nuclear SSU markers, and compared to published sequences using BLAST and BOLD Systems searches. Matches or very near-matches (>99%) for all three of these barcodes alongside an eDNA detection can be considered adequate evidence that C. tumulosa has been identified. However, full genomic sequencing and assembly of the plastidial genome with comparison to NCBI accession number MW309501 may also be performed for a more confident assessment (Paiano et al., 2021).

Conclusions

The alga C. tumulosa is a putative non-native nuisance red alga in its only confirmed geographic location, PMNM, where its spread is likely aided by marine debris. Despite targeted sampling efforts from global herbaria and opportunistic eDNA surveys, there remains no published record of the species’ detection elsewhere, underscoring the challenge of searching across the vast Pacific Ocean. To address this, we suggest a model-guided search strategy to implement in the most probable source locations. These locations are: Japan’s Izu Islands and Cape Inubo, the central Pacific islands of Johnston Atoll and the Line Islands including Palmyra, and in the ETP, Clipperton Atoll and the northern Galápagos archipelago. In short, this protocol begins with eDNA surveys followed by in-water surveys and DNA barcoding. The qPCR assay targeting C. tumulosa is sufficiently sensitive to uncover trace amounts of target eDNA from pooled samples. Through this process we identified a number of Chondria taxa that have yet to be sequenced for any molecular markers, with geographic ranges that include the potential source regions suggested by the oceanographic model. Implementing this systematic and explicit search for C. tumulosa is critical for locating and determining the native range of the species, which would also confirm its non-native status in Hawai‘i—raising it to invasive status. Identification of the native range of the species is ultimately necessary to resolve ecological parameters pertinent to control mechanisms, thereby supporting conservation efforts in PMNM. We note that these methods can also be developed in other cryptogenic marine organisms to help fill in important ecological and evolutionary gaps that go beyond C. tumulosa.

Supplemental Information

Supplemental Information 1 Comparison of diffusivity performance in the backtracking module of the Connectivity Modeling System (CMS) in recreating the origin of tagged marine debris.

The final GPS coordinates for satellite-tagged marine debris objects originating at Manawai, Hawai‘i were designated as the origin for a series of targeted CMS model runs. For each run, 1000 particles were backtracked and their minimum distance to Manawai recorded and displayed using a jitter function alongside the boxplot representing the model run. Each model run varied only in the horizontal diffusivity value provided to CMS at 0.5, 5, 10, 20, and 50 m 2 /s. Mean minimum passing distance (km) to Manawai and significance categories are displayed above their respective diffusivity values. Interquartile ranges (IQRs) for each model run were 97.7-198.0 km (0.5 m 2 /s), 60.7-154.4 km (5 m 2 /s), 46.6-152.2 km (10 m 2 /s), 47.4-159.8 km (20 m 2 /s), and 54.4-181.6 km (50 m 2 /s), with whiskers extending to 1.5*IQR. Dots beyond the whiskers represent outliers. The diffusivity value 10 m 2 /s exhibited the lowest mean passing distance and was selected as the horizontal diffusivity value for the CMS run. Horizontal diffusivities from 5-50 m 2 /s did not vary significantly.

Supplemental Information 2 The number of landings in each hexagonal settlement polygon recording any landings during the Connectivity Modeling System backtracking run.

Landings are colored in accordance with the color key shown in Figure 1. Particles backtracked from Manawai, Hawai‘i were released and tracked from January 1, 2000 to December 30, 2015 until contacting a settlement polygon in the Pacific Ocean. The cutoff between polygons colored purple and light blue is 1% of all settled particles. Most hexagons with landings did not receive many particles while the 18 hexagons of varying colors received many more.

Supplemental Information 3 Maps of the three eDNA sampling regions for Chondria tumulosa during 2024: (A) their locations in the Pacific Ocean; (B) Okinawa, Japan (n = 20); (C) Johnston Atoll (n = 40); and (D) Kiritimati Atoll in the Line Islands (n = 22).

In each region, 2-L duplicate seawater samples were collected from within 1 m of the bottom. To increase screening efficiency, eDNA from replicate samples was pooled within each region and amplified via qPCR. A second pool containing spiked positive eDNA from a known site (Kuaihelani, PMNM) was created to verify assay sensitivity. No sampling locations included a positive hit for C. tumulosa. The red boxes in panel A represent the sampling locations across the Pacific and blue dots in panels B-D represent the individual samples taken. Gray triangle represents the location of Manawai.

Supplemental Information 4 Density clouds of particle locations throughout the modeled Connectivity Modeling System backtracking period to each region of interest.

Legend and colors indicate the proportion of particles located in each position throughout the study period. Particle pathways are shown originating from Japan (A), the central Pacific (B), and the Eastern Tropical Pacific (C) as potential introduction pathways for Chondria tumulosa to Manawai, Hawai‘i. The gray triangle shows the location of Manawai and the box in each panel represents the potential source region.

Supplemental Information 5 Comparison of the time spent drifting in the Connectivity Modeling System backtracking module prior to settlement between regions.

The drift times for particles settling in the central Pacific, Japan, and the eastern Pacific all exhibit significant differences with mean drifting time increasing from left to right. The median (mean) drift times for the central Pacific, Japan, and eastern Pacific were 356 (420), 497 (552), and 950 (1007) days, respectively. Interquartile ranges (IQRs) for each location were 97.7-198.0 days (central Pacific), 60.7-154.4 days (Japan), and 46.6-152.2 days (eastern Pacific), with whiskers extending to 1.5*IQR. Dots beyond the whiskers represent outliers.

Supplemental Information 6 Details of environmental DNA samples screened for the presence of Chondria tumulosa.

Supplemental Information 7 Details of samples compared with Chondria tumulosa.

Included are those generated in this study and those acquired from GenBank, sorted by BLAST similarity to NCBI GenBank accession NC057618, the published plastidial genome of the alga. Identities of specimens reflect the given GenBank name or the original identification of the specimen if generated in this study.

Supplemental Information 8 Range overlap of Chondria species with regions of interest identified by the Connectivity Modeling System output as potential source locations for Chondria tumulosa in the Papahānaumokuākea Marine National Monument, Hawai‘i.

The species column includes those that are either currently accepted taxonomically, have unresolved taxonomic status, or have been transferred to other genera within the family Rhodomelaceae. Availability of DNA barcode sequences for these species from GenBank NCBI is indicated.

Supplemental Information 9 Total modeled particle landings in each polygonal settlement area included in the Connectivity Modeling System backtracking run.

Longitude and latitude represent the centroid of each polygon. Latitude is given in positive and negative values, with those below zero representing degrees south. Longitude is given from 0-360 with those values of western longitude represented by positive numbers. This was necessary for modeling the particles across the meridian.

Supplemental Information 10 Pairwise Wilcoxon rank sum test p-values, with Bonferroni correction, comparing the landings originating in the settlement regions of interest between El Niño, La Niña, and Neutral periods.

Displayed p-values represent those from Japan/The central Pacific/the Eastern Tropical Pacific. Significance is indicated by an asterisk.

Supplemental Information 11 Pairwise Wilcoxon rank sum test p-values, with Bonferroni correction, comparing the landings originating in the settlement regions of interest between seasons.

Displayed p-values represent those from Japan/the central Pacific/the eastern Pacific. Seasons are December, January, and February (DJF); March, April, and May (MAM); June, July, and August (JJA), and September, October, and November (SON).

Supplemental Information 12 Displayed p-values represent those from Japan/the central Pacific/the eastern Pacific. Seasons are December, January, and February (DJF); March, April, and May (MAM); June, July, and August (JJA), and September, October, and November (SON).

Significance is indicated by an asterisk.

Supplemental Information 13 NCBI sequences generated in this study.

Supplemental Information 14 UPA alignment.

Alignment of sequences generated in this study and those of similar taxa. Barcodes are from the Universal Plastid Amplicon (UPA) marker.

We wish to thank members of the Sherwood Algal Biodiversity Lab (S. Paradis, K. Allsopp, S. Geise, L. Reynes, M. Paiano, M. Eggertsen, F. Cabrera, T. Irvine, and V. Velasco), the Papahānaumokuākea Marine Debris Team, colleagues with whom discussions improved the manuscript (B. Bowen, K. McCoy, L. Maki, and B. Sanford), and attendees and organizers of the Chondria Conference “ChonCon” held in May 2024. Phaik-Eem Lin sent samples from the herbarium of the University of Malaya, Kuala Lumpur, Malaysia.

Additional Information and Declarations

Competing Interests

The authors declare that they have no competing interests.

Author Contributions

James T. Fumo conceived and designed the experiments, performed the experiments, analyzed the data, prepared figures and/or tables, authored or reviewed drafts of the article, and approved the final draft.

Patrick K. Nichols conceived and designed the experiments, performed the experiments, analyzed the data, prepared figures and/or tables, authored or reviewed drafts of the article, and approved the final draft.

Taylor Ely conceived and designed the experiments, performed the experiments, analyzed the data, authored or reviewed drafts of the article, and approved the final draft.

Peter B. Marko conceived and designed the experiments, performed the experiments, analyzed the data, authored or reviewed drafts of the article, and approved the final draft.

Amy L. Moran conceived and designed the experiments, performed the experiments, analyzed the data, authored or reviewed drafts of the article, and approved the final draft.

Brian S. Powell conceived and designed the experiments, analyzed the data, authored or reviewed drafts of the article, and approved the final draft.

Taylor M. Williams conceived and designed the experiments, authored or reviewed drafts of the article, and approved the final draft.

Randall K. Kosaki conceived and designed the experiments, authored or reviewed drafts of the article, and approved the final draft.

Celia M. Smith conceived and designed the experiments, authored or reviewed drafts of the article, and approved the final draft.

Keolohilani H. Lopes Jr. conceived and designed the experiments, authored or reviewed drafts of the article, and approved the final draft.

Jennifer E. Smith conceived and designed the experiments, authored or reviewed drafts of the article, and approved the final draft.

Heather L. Spalding conceived and designed the experiments, authored or reviewed drafts of the article, and approved the final draft.

Stacy A. Krueger-Hadfield conceived and designed the experiments, authored or reviewed drafts of the article, and approved the final draft.

Karla J. McDermid conceived and designed the experiments, authored or reviewed drafts of the article, and approved the final draft.

Brian B. Hauk conceived and designed the experiments, performed the experiments, authored or reviewed drafts of the article, and approved the final draft.

James Morioka conceived and designed the experiments, performed the experiments, authored or reviewed drafts of the article, and approved the final draft.

Kevin O’Brien conceived and designed the experiments, performed the experiments, authored or reviewed drafts of the article, and approved the final draft.

Barbara Kennedy conceived and designed the experiments, performed the experiments, authored or reviewed drafts of the article, and approved the final draft.

Frederik Leliaert conceived and designed the experiments, performed the experiments, authored or reviewed drafts of the article, and approved the final draft.

Mutue T. Fujii conceived and designed the experiments, performed the experiments, authored or reviewed drafts of the article, and approved the final draft.

Wendy A. Nelson conceived and designed the experiments, performed the experiments, authored or reviewed drafts of the article, and approved the final draft.

Stefano G. A. Draisma conceived and designed the experiments, performed the experiments, authored or reviewed drafts of the article, and approved the final draft.

Alison R. Sherwood conceived and designed the experiments, performed the experiments, analyzed the data, authored or reviewed drafts of the article, and approved the final draft.

Field Study Permissions

The following information was supplied relating to field study approvals (i.e., approving body and any reference numbers):

Water samples for eDNA analysis were collected under USFWS Special Use Permit 12543-24001 (Johnston Atoll), a Research Consent Certificate from the Ministry of Fisheries and Marine Resources Development (Kiribati), and with the permission of the University of the Ryukyus (Okinawa).

DNA Deposition

The following information was supplied regarding the deposition of DNA sequences:

The DNA barcoding sequences generated in this study are available at GenBank: PV036782–PV036852 (Table S2).

Data Availability

The following information was supplied regarding data availability:

The data and code are available at Zenodo: Fumo, J. (2025). A predictive framework for identifying source populations of non-native marine macroalgae: A case study of Chondria tumulosa in the Pacific Ocean [Data set]. Zenodo. https://doi.org/10.5281/zenodo.14948153.

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
