# Peer review of "A predictive framework for identifying source populations of non-native marine macroalgae: Chondria tumulosa in the Pacific Ocean"

_PeerJ, doi:10.7717/peerj.19610_

## Round 0.1 · original submission · Minor Revisions

We are pleased to let you know that your manuscript has now passed through the review stage and is ready for revision. Many manuscripts require a round of revisions, so this is a normal but important stage of the editorial process. Please edit the manuscript carefully in accordance with the comments of the reviewers. This will help avoid further rounds of explanations and revisions, and allow me to quickly move to the decision.

Reviewer 1 ·

Basic reporting

The manuscript meets the standards of high-quality scientific publication.It's well written. Illustrations are good. I enjoyed reading it. The introduction provides a comprehensive overview of the problem, the hypothesis is well supported, the discussion is very detailed, the proposed sources of Chondria tumulosa and the migration routes of the species from them to Hawaii are well supported by the relevant literature. Although the presence of Chondria tumulosa in Clipperton Atoll, Johnston Atoll, Line Island, the Galapagos Islands, and Japan has not been confirmed by eDNA analysis and extensive herbarium examination, oceanographic modeling results strongly support these areas as potential sources of C. tumulusa in Hawaii. Further studies using the authors' recommended procedure for searching for C. tumulosa in regions of interest may prove more successful. I recommend the manuscript for publication in PeerJ without changes.

Experimental design

The design of the experiment is adequate to answer the questions posed.

Validity of the findings

The results and conclusions are well-founded

·

Basic reporting

This study introduces a novel methodology to address a gap in the investigation of alien marine macroalgae. The authors set out with the ambitious aim of demonstrating that a seemingly endemic alga is, in fact, an alien invasive species. While the study employs multiple approaches, none of them succeed in substantiating the central hypothesis.

The authors appear to be on the right path conceptually, but due to the lack of supporting results, the manuscript ultimately circles back to the ideas already introduced at the beginning. As a result, the study does not convincingly identify the origin of the target alga, and this cannot remain the central focus of the paper in its current form.

Since the authors already possess the necessary samples, one alternative would be to apply their methodology to another species known to occur both in the hypothesised source regions and in Hawai‘i. This would allow them to validate their approach—specifically the combination of oceanographic modelling and eDNA sampling—even if the original case study remains inconclusive. As for the herbarium specimen sequencing, the authors should clearly state that the method did not work in this particular case, at least under the current conditions.

I also recommend that the authors make all datasets and any new code publicly available where possible, or clearly state what is available upon request.

Section-Specific Feedback:

- Introduction: This section currently makes up approximately 25% of the manuscript and should be significantly trimmed. It should focus more directly on background information relevant to this study and clearly state its goals.

- Methods: Some aspects of the methodology require further detail and clarification to ensure reproducibility and to enhance reader understanding.

- Results: This section would benefit from rephrasing for clarity. In particular, the authors should better explain the implications and reasoning behind the results presented.

- Discussion: The discussion contains several unsupported or speculative claims. Since the central hypothesis remains unproven—i.e., it is still unknown whether this apparently endemic species is alien—the authors should shift their focus to the value and potential of their methodology in algal research. At present, the discussion on modelled connectivity is not justified, as no concrete connections were established. Testing the approach on a case with a known positive result would greatly strengthen the manuscript and showcase its utility.

Experimental design

Since the target species was not detected in environmental samples from the proposed source regions, it remains unclear whether this outcome reflects true absence or false negatives due to limitations in sampling, detection sensitivity, or eDNA degradation. This uncertainty weakens the ability to draw definitive conclusions about species distribution and connectivity.

Given that the authors already have access to environmental samples from both the source regions and Hawai‘i, I strongly recommend testing these samples for other marine macroalgal species that are known to occur in both locations. Successfully detecting such species would help validate the effectiveness of their methodology (i.e., eDNA sampling combined with oceanographic modeling) and demonstrate its broader applicability—even if the primary case study remains inconclusive.

Validity of the findings

The primary strength of this study lies in its methodological innovation, aiming to fill a clear gap in algal research by combining eDNA analysis with oceanographic modeling. However, the current version of the manuscript places significant emphasis on interpreting connectivity results, despite the fact that no direct or confirmed connection was observed between the source regions and the study site.

Without positive evidence of connectivity—either through species detection or tracking data—these interpretations remain speculative. The authors should refocus the discussion on the utility, limitations, and future potential of their methodology, rather than drawing conclusions about biogeographic links that are not supported by the data.

Additional comments

In several instances, the manuscript relies on outdated, general, or fauna-focused references. Given the specific focus on marine macroalgae, more recent and macroalgae-relevant literature should be consulted and cited where appropriate. I have provided examples from my own publications as a starting point, but I strongly encourage the authors to perform a broader search for up-to-date, field-specific references.

For a more thorough review, I have attached a PDF with detailed questions, suggestions, and comments directly on the manuscript. These should be carefully addressed in the revision process.

·

Basic reporting

The manuscript is very well-written in clear and professional English throughout, and is appropriate for a scholarly audience. Minor edits for clarity or to fix typos/grammar will improve readability in a few places; I have highlighted and provided feedback on these directly in the PDF.

The introduction effectively sets up the research question and provides a great deal of useful background for readers to understand the significance of the study. The context is well established and aligns with the scope of the journal.

The manuscript demonstrates impressive engagement with the relevant literature. Citations are appropriate and up to date. I did not find any content that was unsupported by reference material or lacking in relevant sources.

The overall structure of the paper follows standard conventions for the discipline and aligns well with journal expectations. The subheadings under longer sections improves organization and helps guide the reader through the authors’ rationale, methods and findings.

Figures and tables are appropriate and contribute meaningfully to the presentation of results. They are clearly labelled, described in the text, and visually appear to be of high quality. My only comment is to carefully consider the text color in the Figure 2 current names. Where the black text appears against a dark color (e.g., purple, blue) on the density cloud they can be difficult to read.

The raw data are supplied and appear to be complete.

Experimental design

This study presents original primary research that falls well within the scope of the journal. The research question is clearly articulated, relevant, and addresses a timely and meaningful topic in marine ecology and biogeography. The authors explicitly state how their work fills a knowledge gap concerning the origin and potential dispersal of a putatively non-native alga.

The investigation incorporates a multidisciplinary approach, combining oceanographic modeling, eDNA sampling and analysis, and DNA barcoding. The integration of these tools allows for a comprehensive exploration of potential dispersal pathways and source populations.

While I am less familiar with the specific oceanographic modeling techniques used, the rationale for their inclusion is sound, and the approach appears appropriate for the study objectives.

The eDNA sampling and analysis followed standard protocols and appeared to be well controlled. From what is presented, the procedures were robust and the interpretation of results was handled carefully.

I am more familiar with the DNA barcoding component of the study, and I found these methods to be rigorous and well-executed. The authors used a broad comparative framework, including herbarium material from multiple regions. The protocols used are well-established in the literature. I was curious about the suggestion to use a modified CTAB protocol for DNA extraction and wonder if the authors might clarify whether this is recommended for archival herbarium specimens only or also for specimens that have been preserved in silica or LN2 shortly following collection, and have noted this question in my comments on the PDF.

Overall, the experimental design is rigorous and adheres to high technical and ethical standards. The methods are described with enough detail and/or references to allow replication, and the combination of complementary tools enhances the strength of the findings.

Validity of the findings

While the authors were not able to validate the model-predicted source locations using eDNA evidence, or match the DNA of the putative non-native alga to specimens collected outside of the PMNM or from herbaria, their conclusions are appropriately cautious and clearly linked to the original research question. The limitations of the findings are acknowledged transparently, and the interpretation remains grounded in the data.

All underlying data are provided and appear to be robust, well controlled, and analyzed with appropriate statistical and methodological rigor. The use of multiple complementary approaches—oceanographic modeling, eDNA analysis, and DNA barcoding using multiple genetic markers—enhances the credibility of the study, even in the absence of a definitive match.

Importantly, the authors present a compelling case for the value of this multidisciplinary framework in efforts to trace the origins of potential marine invaders. Their discussion highlights how future studies might build on this approach, and they provide a strong rationale for the continued development and replication of this work, particularly given the ecological importance of early detection and source identification.

In sum, while the central hypothesis remains unresolved, the study makes a meaningful methodological contribution, and the conclusions are appropriately limited to what the data can support.

Additional comments

This was a thoroughly engaging manuscript. I genuinely enjoyed reading it and found the research to be both compelling and highly relevant to ongoing efforts to understand species introductions and dispersal in marine ecosystems. Although some of the key findings remain inconclusive, the authors present a strong and well-reasoned case for the value of their interdisciplinary approach, and lay important groundwork for future investigations.

I did note the first author is a PhD student who very recently defended their thesis! (Congratulations!) I appreciate the clarity of the writing, the ambition and scope of the study, and I was impressed by the thoughtful integration of diverse methods. It’s exciting to see this kind of work being done, and I look forward to seeing how this research program continues to develop!

---

## Round 0.2 · accepted · Accept

In the revised version the authors took into consideration all comments and remarks. I recommend to accept the manuscript for publication in PeerJ.